# Improved Left Atrial Function in CRT Responders: A Systematic Review and Meta-Analysis

**DOI:** 10.3390/jcm9020298

**Published:** 2020-01-21

**Authors:** Ibadete Bytyçi, Gani Bajraktari, Per Lindqvist, Michael Y. Henein

**Affiliations:** 1Institute of Public Health and Clinical Medicine, Umeå University, 90187 Umeå, Sweden; i.bytyci@hotmail.com (I.B.); gani.bajraktari@umu.se (G.B.); per.lindqvist@umu.se (P.L.); 2Clinic of Cardiology, University Clinical Centre of Kosovo, 10000 Prishtina, Kosovo; 3Faculty of Medicine, Department of Surgical and Perioperative Sciences, Clinical Physiology, Umeå University, 90187 Umeå, Sweden; 4Molecular and Clinic Research Institute, St George University, London SW17 0QT, UK

**Keywords:** left atrial strain, cardiac resynchronization therapy, heart failure

## Abstract

Cardiac resynchronization therapy (CRT) is associated with reverse left atrial (LA) remodeling. The aim of this meta-analysis was to assess the relationship between clinical response to CRT and LA function changes. We conducted a systematic search of all electronic databases up to September 2019 which identified 488 patients from seven studies. At (mean) 6 months follow-up, LA systolic strain and emptying fraction (EF) were increased in CRT responders, with a −5.70% weighted mean difference (WMD) [95% confidence interval (CI) −8.37 to −3.04, *p* < 0.001 and a WMD of −8.98% [CI −15.1 to −2.84, *p* = 0.004], compared to non-responders. The increase in LA strain was associated with a fall in left ventricle (LV) end-systolic volume (LVESV) *r* = −0.56 (CI −0.68 to −0.40, *p* < 0.001) and an increase in the LV ejection fraction (LVEF) *r* = 0.58 (CI 0.42 to 0.69, *p* < 0.001). The increase in LA EF correlated with the fall in LVESV *r* = −0.51 (CI −0.63 to −0.36, *p* < 0.001) and the increase in the LVEF *r* = 0.48 (CI 0.33 to 0.61, *p* = 0.002). The increase in LA strain correlated with the increase in the LA EF, *r* = 0.57 (CI 0.43 to 0.70, *p* < 0.001). Thus, the improvement of LA function in CRT responders reflects LA reverse remodeling and is related to its ventricular counterpart.

## 1. Introduction

Heart failure (HF) is a clinical syndrome that is becoming a major public health problem worldwide because of its increasing incidence and prevalence as well as its related morbidity and mortality [1]. Despite its failure in about one-third of treated patients, cardiac resynchronization therapy (CRT) still remains the best treatment for symptomatic HF patients on full medical therapy [2,3].

Responders to CRT have shown clear evidence for improved cardiac performance, left ventricular (LV) function, and the reverse remodeling of the left atrium (LA) [4,5]. LA function can be assessed by various echocardiographic techniques, among which myocardial deformation has recently shown significant accuracy. Despite this, the relationship between CRT-related LA and ventricular function changes remains poorly established, irrespective of the fact that CRT is associated with both cavity reverse remodeling and reduced atrial arrhythmia [6,7,8].

The aim of this meta-analysis was to assess the relationship between clinical response to CRT, LA function improvement, and LV function improvement.

## 2. Methods

The research methodology used in this study followed the Meta-analysis of Observational Studies in Epidemiology (MOOSE) statement for reporting systematic reviews and meta-analyses of observational studies [9]. Due to the study design (meta-analysis), neither Institutional Review Board (IRB) approval nor informed consent was needed [10].

### 2.1. Search Strategy

We systematically searched PubMed–Medline, EMBASE, Scopus, Google Scholar, the Cochrane Central Registry of Controlled Trials, and ClinicalTrial.gov, up to September 2019, with the following key words: “Cardiac resynchronization therapy” OR “CRT” AND “Left atrial function” OR “LA function” OR “Left atrial strain” OR “LA strain” OR “LA emptying fraction” OR “LA ejection fraction (EF)” AND “Outcome” OR “CRT responders” OR “CRT non-responders” AND “Follow-up”.

Additional searches for potential trials included the references of review articles on the subject and the abstracts presented at the scientific sessions of the European Society of Cardiology (ESC), the American Heart Association (AHA), American College of Cardiology (ACC), and European Association of Cardiovascular Imaging (EACVI). The wild-card term “*” was used to enhance the sensitivity of the search strategy. The literature search was limited to articles published in English and to studies of humans.

Two reviewers (I.B and G.B) independently evaluated each article. No filters were applied. The remaining articles were obtained in full-text and assessed, again by the same two researchers who separately evaluated each article and carried out data extraction and quality assessment. Disagreements between the reviewers were resolved by discussion with a third party (M.Y.H.).

### 2.2. Study Selection

The criteria for inclusion in the meta-analysis were: (i) studies investigating patients undergoing cardiac resynchronization therapy; (ii) reporting left atrial predictors of CRT responders and non-responders; (iii) over three months of completed follow-up; and (iv) an enrolled population of adults aged ≥18 years.

Exclusion criteria were: (i) insufficient statistical data to compare two groups; (ii) less than three months follow-up period; (iii) non-human subjects; and (iv) articles not published in English.

### 2.3. Outcome Variables

The key clinical end-points were the relationships between clinical response to CRT and LA function changes. The main outcome measures were LA function, peak atrial longitudinal strain (PALS), and the total EF.

### 2.4. Data Extraction

Eligible studies were reviewed, and the following data were abstracted: (1) first author′s name; (2) year of publication; (3) study design; (4) data on two arms: CRT responders and non-responders; (5) LA strain measured by echocardiography; (6) baseline characteristics of the patients; (7) baseline LA strain; (8) mean follow-up period; (9) age and gender of study participants; and (10) follow-up LA function.

Peak atrial longitudinal strain (PALS) was measured as the first peak between the QRS complex and the T wave during LV systole, guided by the superimposed electrocardiogram (ECG). When P–P gating was used, and the strain curve was taken as the negative deflection (that represents the peak LA contraction strain) followed by a positive deflection. The sum of the two parts of the curve (negative + positive) represented the peak LA wall strain (Figure 1) [11]. The two types of gating were extracted from the papers’ available text or images.

### 2.5. Quality Assessment

The assessment of risk of bias and applicability concerns in the included studies was evaluated by the same investigators by using the quality assessment of diagnostic accuracy studies questionnaire (QUADAS-2), which was optimized for these study questions (Appendix A) [12]. The QUADAS-2 tool has 4 domains for risk of bias-patient selection, index test, reference test, and flow and timing-and three domains for applicability-patient selection, index, and reference test domains.

### 2.6. Statistical Analysis

The meta-analysis was conducted with statistical analysis that was performed with the RevMan software (Review Manager Version 5.1, The Cochrane Collaboration, Copenhagen, Denmark), with two-tailed *p* < 0.05 considered as significant. Weighted mean differences (WMD) and a 95% confidence interval (CI) were calculated for each study. The baseline characteristics are reported as the median and range. Mean and standard deviation (SD) values were estimated by using the method described by Hozo et al. [13].

To test the potential associations between LA function and CRT response, we used the MedCalc program (Version 19.0, Medcalc Sotware, Ostend, Belgium) and applied the Hedges–Olkin (1985) method for calculating the weighted summary correlation coefficient under the fixed/random effects model by using a fisher Z transformation of the correlation coefficients. The heterogeneity statistics were incorporated to calculate the summary correlation coefficient under the random effects model (DerSimonian and Laird, 1986).

The meta-analysis is presented in forest plots and was performed with a fixed-effects model. The heterogeneity between studies was assessed with Cochran’s Q test and the *I*^2^ index. As a guide, *I*^2^ < 25% indicated a low heterogeneity, 25–50% indicated a moderate heterogeneity, and >50% a indicated high heterogeneity [14]. To assess the additive (between-study) component of variance, the reduced maximum likelihood method (*tau*^2^) incorporated the occurrence of residual heterogeneity into the analysis [15]. Publication bias was assessed via visual inspections of funnel plots and Egger’s test.

## 3. Results

### 3.1. Search Results and Trial Flow

Of 2819 articles identified in the initial search, 201 were screened as potentially relevant. After excluding 201 studies, 20 full articles were assessed according to the inclusion and exclusion criteria. After careful assessment of these 20 articles, 13 were excluded, and only seven articles were included in the final analysis [16,17,18,19,20,21,22] (Appendix A). In all LA function studies, two dimensional (2D) LA strain (PALS) was measured based on R–R gating, and the sonographers were blinded to pressure measurements. (Appendix A).

### 3.2. Characteristics of Included Studies

A total of 488 patients from the seven observational studies were included (Table 1). CRT responders were 181 and CRT non-responders were 118, with mean follow-up period of six months for both groups. The mean age of patients was 62.1 ± 10.2 years, and 68.3% were males. The mean QRS duration was 156 ± 38, and ischemic etiology for heart failure was found in 33.4% of patients. Between the two groups of patients, CRT responders and non-responders had no difference in age (61.9 ± 7.3 vs. 62.4 ± 10 years, *p* = 0.67, respectively), male gender (69% vs. 66%, *p* = 0.19), ischemic etiology (34% vs. 31.8%, *p* = 0.58) or QRS duration (155.2 ± 31 vs. 156.3 ± 34 ms, *p* = 0.42, Table 2).

### 3.3. LA Function in CRT Responders Versus CRT Non-Responders

The pooled analysis showed that CRT responders had no baseline difference in LA strain compared to non-responders, with a WMD of 1.46% [95% CI from −1.58 to 4.50, *p* = 0.35], whereas the LA EF was higher with a WMD of 4.25% [95% CI from 0.42 to 8.08, *p* = 0.03; Figure 2a,b] in responders.

At follow-up, LA strain increased significantly in CRT responders with a WMD of −5.70% [95% CI from −8.37 to −3.04, *p* < 0.001] compared to non-responders, in whom it remained unchanged, with a WMD of 1.29% [95% CI from −2.08 to 4.67, *p* = 0.45; Figure 3a,b]. Likewise, the LA EF increased in CRT responders: WMD = −8.98% [95% CI from −15.1 to −2.84, *p* = 0.004] vs. WMD = −0.50% [95% CI from −13.3 to 12.3, *p* = 0.10] in non-responders (Figure 4). Heterogeneity across the included studies was not encountered at follow-up in either CRT responders or non-responders (Chi^2^ = 4.05, I^2^ = 26 df = 3, and *p* = 0.60 vs. Chi^2^ = 4.78, I^2^ = 37, df = 3, *p* = 0.37, respectively) except for the moderate heterogeneity detected at the baseline LA strain between the two groups, as tested by the random-effect analysis (Chi^2^ = 7.47, I^2^ = 46, df = 4, *p* = 0.11).

### 3.4. LV Dimension and Function in CRT Responders Versus CRT Non-Responders

There was no difference in baseline LV dimensions, systolic function, and QRS duration between CRT responders and the CRT non-responders: WMD = −2.18% [95% CI from −24.01 to 19.6, *p* < 0.85]; baseline left ventricle end-systolic volume (LVESV): WMD = −4.22% [95% CI from −22.1 to 13.6, *p* = 0.64]; baseline left ventricular end-diastolic dimension - LVEDd: WMD = −1.74% [95% CI from −4.76 to 1.27, *p* = 0.26]; baseline left ventricle ejection fraction (LVEF): WMD = 0.76% [95% CI from −3.34 to 4.86, *p* = 0.72] (Appendix A); and baseline QRS duration; WMD = −2.60% [95% CI from −10.9 to 5.79, *p* = 0.54] (Appendix A). Similarly, no difference was found in the LA dimension, which had a WMD of −2.17% [95% CI from −7.1 to 2.67, *p* = 0.38], between the two groups.

### 3.5. The Relationship between LA and LV Function in CRT Responders

To test for potential associates with CRT response, we calculated the weighted summary correlation coefficient between the LA function and LV parameters of CRT responders. This analysis showed that the increase in LA strain was associated with a fall in left ventricular end-systolic volume - LVESV’s weighted summary correlation (*r*) [*r* = −0.56 (CI from −0.68 to −0.40, *p* < 0.001) Q^2^ = 0.06, df = 0.2, I^2^ = 0.0%, *p* = 0.86] and an increase in the LVEF [*r* = 0.58 (CI from 0.42 to 0.69, *p* < 0.001) Q^2^ = 0.72, df = 2, I^2^ = 0.0%, *p* = 0.69; Figure 5,b]. Similarly, although with a less significance, the increase in the LA EF correlated with the fall in LVESV [*r* = −0.51 (CI from −0.63 to −0.36, *p* < 0.001) Q^2^ = 0.92, df = 0.2, I^2^ = 0.0%, *p* = 0.86] and with the increase in the LVEF [*r* = 0.48 (CI from 0.33 to 0.61, *p* = 0.002) Q^2^ = 0.36, df = 0.2, I^2^ = 0.0%, *p* = 0.83; Figure 5c,d]. The increase in LA strain correlated with the increase in the LA EF (r = 0.57 (CI from 0.43 to 0.70, *p* < 0.001) Q2 = 0.16, df = 1, I2 = 0.0%, *p* = 0.64; Appendix A).

### 3.6. Relationship between LA Strain Change and Baseline Age and Male Gender

The % mean change of LA strain was not related to baseline mean age or male gender (β = −0.20 (from 0.159 to −0.516), *p* = 0.20, Tau = 0.00%, I^2^ = 0.00%, Q = 0.19, d = 4) or age(β = −0.24 (from 0.22 to −0.69), *p* = 0.27, Tau = 0.00% I^2^ = 0.00%, Q = 1.20, d = 4) (Appendix A). There was no heterogeneity across the included studies.

### 3.7. Risk of Bias Assessment

Based on quality assessment of diagnostic accuracy studies questionnaire (QUADAS-2), four domains of criteria for risk of bias and three for applicability were analyzed, and the risk of bias was assessed as “low risk”, “high risk”, or “unclear risk” (Appendix A) [10]. Most studies had a low or moderate risk of bias and clearly defined their objectives and the main outcomes (Appendix A). The QUADAS-2 analysis for bias evaluation showed all domains to have had low risk of bias (≤40%). Additionally, there was no evidence for publication bias, as evaluated by Egger’s test for our findings.

## 4. Discussion

Despite its failure in about one-third of treated patients, cardiac resynchronization therapy (CRT) still remains the best treatment for symptomatic heart failure patients on full medical therapy, as stated in the European and American guidelines [23,24]. One of the known causes of symptoms in such patients is atrial arrhythmia, which is known to be related to LA cavity enlargement and disturbed function, both of which may improve in CRT responders [16,18]. The regression of atrial arrhythmia with CRT treatment has been reported and interpreted on the basis of reversed LA cavity remodeling [25,26]. Despite these suggestions, the exact contribution of LA function in cardiac reverse remodeling related to CRT remains poorly established [27]. This meta-analysis evaluated the relationship between LA function in patients who received CRT for heart failure.

*Findings:* Our analysis shows that CRT responders had no baseline difference in LA strain compared CRT non-responders, but the LA EF was higher in responders. At follow-up, both LA strain and the LA EF only significantly increased in responders. The increase in LA strain was associated with a fall in LVESV and a rise in the LVEF. Similarly, although with a less significance, the increase in the LA EF correlated with the fall in LVESV and the increase in the LVEF. Finally, the increase in LA strain correlated with the increase in the LA EF.

*Data Interpretation:* LA function is an integral part of cardiac function, and its pump function normally contributes by at least one third to overall LV filling, which increases with age [28]. In heart failure, responders to CRT are mainly those with worse LV dyssynchrony, which itself compromises LA emptying and consequently stroke volume. Additionally, reduced LA emptying results in raised LA pressure and, consequently, myocardial stretch, which then leads to cavity function instability and arrhythmia. Studies have shown that an increase in LA volume is the most accurate predictor of atrial arrhythmia [29,30].

With an optimum response to CRT, LV systolic function improves and the ejection fraction increases, and ESV falls as stroke volume increases. These changes have been shown to have significant hemodynamic effects on overall cardiac performance and symptoms [30]. Furthermore, an improved LV pump function results in better LA emptying as a further contribution to stroke volume. The other side of the benefit from CRT lies at the LA myocardial function level; as has been shown by our results, LA strain increases parallel to the increased cavity emptying fraction, thus providing further evidence for the improvement in LA integral function in the form of myocardial intrinsic function and overall pump performance. It is of interest that these aspects of improvements of LA function did not happen independently of the LV, but they were associated, although modestly, with the fall in LVESV and the increase in ejection fraction [31,32,33]. Such a relationship is of clinical and academic interest, since the described fall in LVESV, usually referred to as a sign of LV reverse remodeling, seems to happen also in a similar way in the left atrium, with an increase in emptying fraction [20,34]. Thus, although the term reverse remodeling might sound non-specific, our results shed light on some of its ingredients in the setting of LV and LA structure and function changes in response to CRT [15,16,17,18,19,20,21]. Finally, our results highlight the fact that the LA is not only a conduit chamber but also a more complex anatomical and function structure, both in and of itself and in its relationship with the LV [35,36]. Our recently published meta-analysis [5] showed a concordant relationship between LA indexed volume and LV volume and function with CRT. The documented improvement of LA myocardial intrinsic function and emptying function is expected to be associated with a fall in cavity pressure with its direct implications on the frequency of atrial arrhythmias known in patients with significant heart failure [11].

*Limitations:* The analysis of LA function and LVESV and/or LVES was based on a small number of studies, so its results should be seen as having modest accuracy until proven in a larger number of studies. The data included in the meta-analysis were collected from the published papers, on whose quality we did not have control; we had to trust the academic merit of the investigators. We were unable to comment on the relevance of our findings in controlling atrial arrhythmia in the analyzed studies because of the limited available data. Likewise, we had hoped to provide evidence for long term benefits from CRT, but, again, such information was not available in the analyzed studies. Finally, we sought to assess the relationship between the individual and combined LA and LV function changes with CRT against symptoms in more detail, but the available data on LA function parameters that could be used in such analysis were very limited. It would have been of great interest to analyze subgroups of patients according to the concordant/discordant relationships between improvements of LA and LV functions, but, again, such data were not uniformly available in the small studies we analyzed.

*Clinical Implications:* The left atrium is an integral component of the overall cardiac structure and function, and it should be seen more than just a conduit. Based on the anatomical myocardial fiber architecture, the association between the left atrial and left ventricular function changes further strengthen such relationship, particularly in the setting of HF with a reduced EF and increased diastolic pressures. Our findings may assist in explaining the well documented lack of symptomatic improvement with CRT in patients with atrial fibrillation, since significant components of the mechanisms of LA emptying and myocardial contraction that contribute to the overall cavity strain does not exist [37].

## 5. Conclusions

Clinical response to CRT is associated with an improvement of LA function, reflecting cavity reverse remodeling. These changes are related to their ventricular counterparts, thus supporting the importance of assessing LA function in patients treated by CRT for heart failure. Future studies should focus on concordant changes in LA and LV function that contribute to clinical improvement of patients receiving CRT for heart failure.

## Figures and Tables

**Figure 1 jcm-09-00298-f001:**
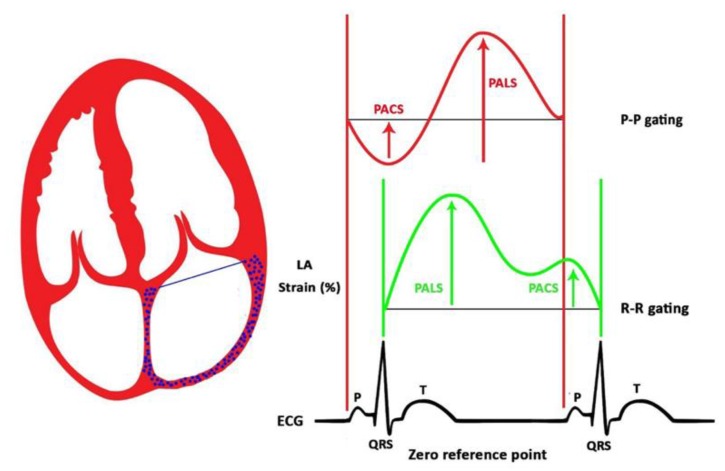
Two types of gating, based on zero reference point (P–P and R–R gating).

**Figure 2 jcm-09-00298-f002:**
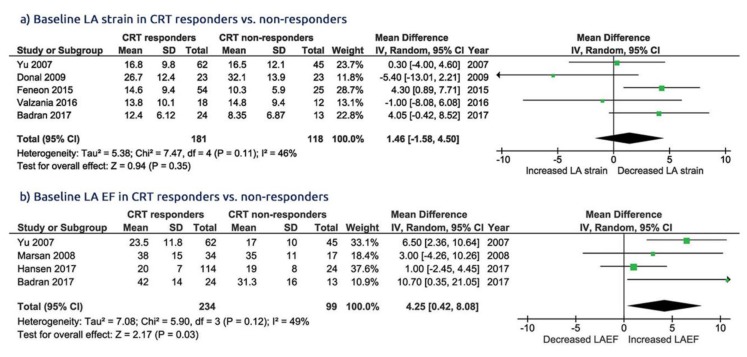
Baseline comparison of LA (left atrial), CRT (function in group of patients with cardiac resynchronization therapy), responders vs. CRT non responders. (**a**) LA strain; (**b**) LA ejection fraction (EF).

**Figure 3 jcm-09-00298-f003:**
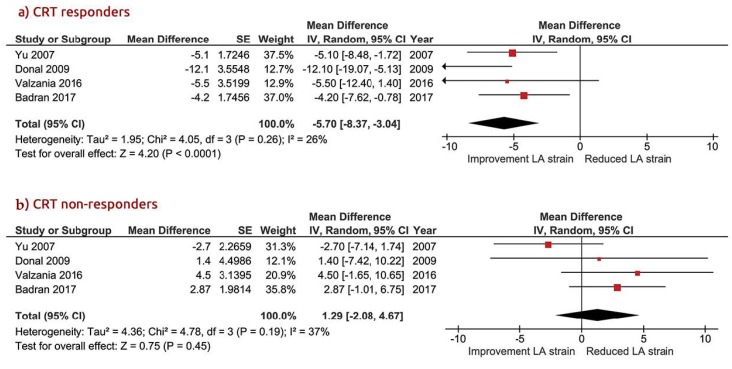
Mean changed LA (left atrial) strain in patients with CRT (cardiac resynchronization therapy). (**a**) CRT responders; (**b**) CRT non responders.

**Figure 4 jcm-09-00298-f004:**
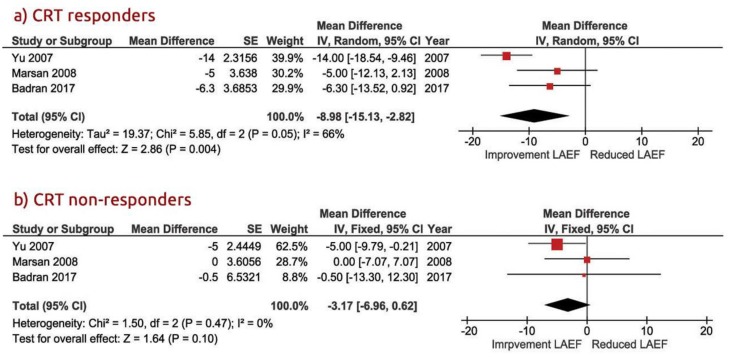
Mean changed LA (left atrial) EF (ejection fraction) in patients with CRT (cardiac resynchronization therapy). (**a**) CRT responders; (**b**) CRT non-responders.

**Figure 5 jcm-09-00298-f005:**
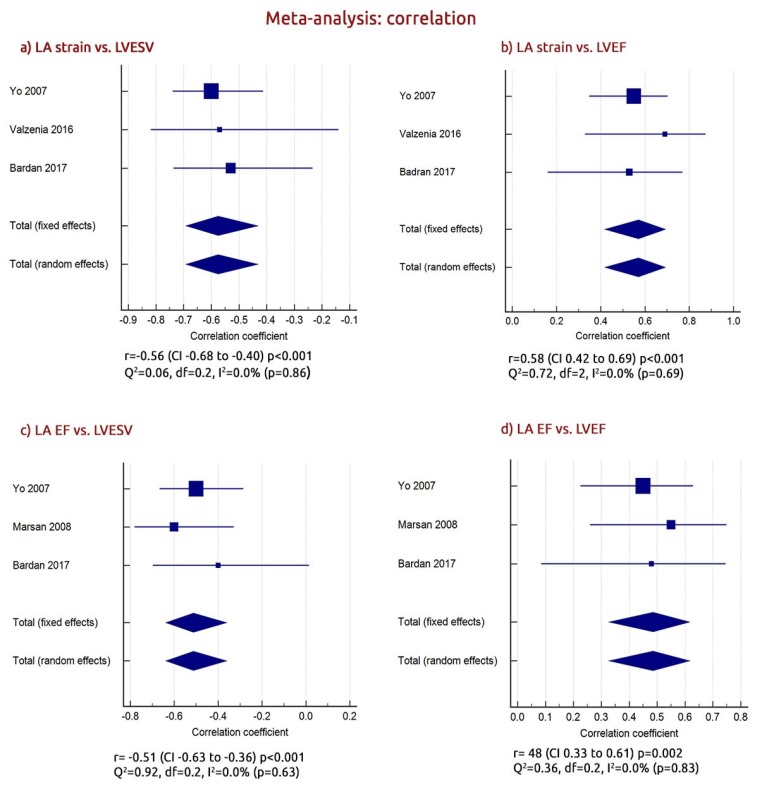
Weighted summary correlation between (**a**) LA (left atrial) ( strain vs. left ventricle end-systolic volume (LVESV); (**b**) LA strain vs. left ventricle ejection fraction (LVEF); (**c**) LA EF (ejection fraction) (vs. LVESV; and (**d**) LA EF vs. LVEF (left ventricle ejection fraction).

**Table 1 jcm-09-00298-t001:** Main characteristics of studies included in the study.

Study, Year	Study	Type of	Inclusion	Exclusion	Key	Echo-	Criteria for	Follow
	Design	Intervention	Criteria	Criteria	Endpoints	Cardiography	CRT Respond	Up
Yo et al. 2007	Prospective	CRT	LV < 40%	Patients with	LA	2DE	LVESV ≥ 15%	3 mo
	observational		QRS ≥ 120 ms	AF	predictors			
			NYHA-III-IV					
Marsan et al. 2008	Prospective	CRT-D	LV ≤ 35%	Patients with	LA and LV	3DE	LVESV ≥ 15%	6 mo
	observational		QRS ≥ 120 ms	AF	predictors			
			NYHA-III-IV					
Donal et al. 2009	Prospective	CRT	HFrEF	Patients with AF	LA	2DE	LVESV ≥ 15%	6 mo
	observational		LV ≤ 35%	Fibrillation; MR	predictors			
			QRS ≥ 120 ms	(EROA > 20 mm^2^)				
Feneon et al. 2015	Prospective	CRT	LV ≤ 35%	Patients with	LA	2DE	LVESV ≥ 15%	6 mo
	observational		QRS ≥ 120 ms	AF	predictors			
			NYHA-II-IV					
			NYHA-III-IV					
Valzania et al. 2016	Prospective	CRT	HFrEF	Patients with	LA	2DE	LVESV ≥ 15%	12 mo
	observational		LV ≤ 35%	AF	predictors			
			QRS ≥ 120 ms					
Badran et al. 2017	Prospective	CRT	LV ≤ 35%	Patients with	LA	2DE	LVESV ≥ 15%	3 mo
	observational		QRS ≥ 120 ms	AF	predictors			
			NYHA-II-IV					
Hansen et al. 2017	Clinical	CRT	LV ≤ 35%	Recently MI	LA	2DE	LVESV ≥ 15%	6 mo
	trial		QRS ≥ 120 ms	CRF, contrast	predictors			
			NYHA-II-IV	allergy				

HF (heart failure), HFrEF (heart failure with reduced ejection fraction), CRT (cardiac resynchronization therapy), LV (left ventricle), EF (ejection fraction), AF (atrial fibrillation), CRF (chronic renal failure), MR (mitral regurgitation), 2DE (two dimensional echocardiography), LVESV (left ventricle end-systolic volume) and mo (months).

**Table 2 jcm-09-00298-t002:** Main characteristics of patients among trials included in the study.

Study, Year	Arms	No.	Age	Male	QRS	NYHA	Ischemic	Mean	Mean
			Year	(%)	Duration (ms)	Functional Class	Etiology (%)	Change of LA Strain %	Change of LA EF %
Yo et al. 2007	R	62	66 ± 11 *	75 *	142 ± 28	3.0 ± 0.5	NR	−5.1	−14
	Non-R	45			154 ± 31	3.0 ± 0.4	NR	−2.7	−5.2
Marsan et al. 2008	R	34	65 ± 7	78	142 ± 28	3.0 ± 0.5	NR	−6	−5.0
	Non-R	17	67 ± 10	70	154 ± 31	3.0 ± 0.4	NR	0	0
Donal et al. 2009	R	23	67 ± 10.4 *	76 *	NR	3.2 ± 0.6 *	NR	−12.1	NR
	Non-R	23						1.4	NR
Feneon et al. 2015	R	54	62.3 ± 10	63	163 ± 27	N−II = 24%	18.6	NR	NR
	Non-R	25	66.5 ± 10	80	158 ± 30	N−II = 22%	60	NR	NR
Valzania et al. 2016	R	18	61 ± 13	63	160 ± 24	2.9 ± 0.2	20	−5.5	−72
	Non-R	12	67 ± 8	50	159 ± 23	3.1 ± 0.3	50	4.5	NR
Badran et al. 2017	R	24	56 ± 9.8	71	NR	N−IV = 33%	29	−4.2	−35.2
	Non-R	13	53 ± 9.5	69	NR	N−IV = 46%	23	2.87	−0.3
Hansen et al. 2017	R	114	69.4 ± 9 *	80 *	166.2 ± 23.0 *	N−IV = 3% *	50 *	−4.4	−5.0
	Non-R	24						−2	NR

R (respond), Non-R (non-respond), LVESV (left ventricle endsystolic volume), LVEF (left ventricle ejection fraction), NR (non-reported) and * only whole group represented. Mean change of LVESV. LVEF was represented only in CRT responders.

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
