# Peer review of "Improved Left Atrial Function in CRT Responders: A Systematic Review and Meta-Analysis"

_jcm, 2020, doi:10.3390/jcm9020298_

Round 1

Reviewer 1 Report

Revision to Bytyci Iet al. of the manuscriptJCM-685859 entitled“Improved left atrial function in CRT responders: a systematic review and  meta-analysis”

In this metanalysis the Authors investigated the relationship between clinical response to CRT and LA function changes.

After a screening of 2819 records only 7 observational studies were included. The Authors found that, patients responders to CRT had greater improvement in LA strain and emptying fraction (EF)  than non-responders. Furthermore these improvements in LA function were correlated with the improvement in LVESV and LVEF. They conclude that the response to CRT involves also LA function and this leads to benefits on the overall cardiac pump function.

Major comments:

TheAuthors state in the Results that after the screening of the articles only 5 were included however from the Tables it is clear that the correct number is 7. Correct this inconsistency The references 21-22 are missing in the text. Most probably this is because in the Results the Authors considered only 5 articles ( ref 16-20) instead of 7 (ref 16.22). Please correct it. In the discussion section my suggestion is to accurately describe the literature about the relation between CRT and LA function.

Minore comments:

Many typos: Abstract: Missing “]” at line 19

           Correct “reserve remodelling” line 26

Figure 2: Correct “strian” Supplement 2: Correct “Incomplite” and “resposers vs responsers” Supplement 5: Correct “ respondrers” Do not use “patients with CRT responds vs no responds”. Use CRT responders vs non responders

Author Response

Reviewer 1

In this metanalysis the Authors investigated the relationship between clinical response to CRT and LA function changes.

After a screening of 2819 records only 7 observational studies were included. The Authors found that, patients responders to CRT had greater improvement in LA strain and emptying fraction (EF) than non-responders. Furthermore, these improvements in LA function were correlated with the improvement in LVESV and LVEF. They conclude that the response to CRT involves also LA function and this leads to benefits on the overall cardiac pump function.

Major comments:

1.The Authors state in the Results that after the screening of the articles only 5 were included however from the Tables it is clear that the correct number is 7. Correct this inconsistency.

Response: We apologize for this mistake. We have corrected this in the revised manuscript.

The references 21-22 are missing in the text. Most probably this is because in the Results the Authors considered only 5 articles ( ref 16-20) instead of 7 (ref 16.22). Please correct it.

Response: Thank you. We have corrected this in the revised manuscript.

In the discussion section my suggestion is to accurately describe the literature about the relation between CRT and LA function.

Response: Thank you. We have now described better the specifically related literature.

Minor comments:

1.Many typos: Abstract: Missing “]” at line 19. Correct “reserve remodelling” line 26.

Response: Thank you. We have now corrected this mistake.

Figure 2: Correct “strian” Supplement 2: Correct “Incomplite” and “resposers vs responsers” Supplement 5: Correct “ respondrers” Do not use “patients with CRT responds vs no responds”. Use CRT responders vs non responders

Response: Thank you for the suggestion. We have now corrected this point.

Reviewer 2 Report

I read with great interest this meta analysis on left atrial reverse remodeling in responders to cardiac resynchronisation therapy (CRT).

Unfortunately, only retrospective analyses could be included and no data from the landmark trials even though they have studied atrial remodelling in some of these databases (but not atrial strain imaging).

I have some more specific comments:

My main problem with the study is that it is mostly descriptive: no associations with atrial arrhythmias or outcome were studied. The focus on CRT responders is in my view intrinsically wrong. There is a large group of patients with discordant remodelling (i.e. left atrial reverse remodelling in CRT non-responders) with improved outcome (Mathias et al. JACC 2016, Kloosterman et al. Europace 2016). The conclusions are unclear to me: if LA remodelling is related to LV remodelling why does it need to be assessed? The most interesting group would be the atrial non-responders in the LV responder group. Hypothetically, in this group remodelling has exceeded the "point of no return". The manuscript needs another round of editing to correct grammatical/semantical mistakes (e.g. LA reserve remodelling instead of reverse remodelling in the first sentence of the conclusions.

Author Response

Reviewer 2

I read with great interest this meta-analysis on left atrial reverse remodeling in responders to cardiac resynchronization therapy (CRT).

Unfortunately, only retrospective analyses could be included and no data from the landmark trials even though they have studied atrial remodelling in some of these databases (but not atrial strain imaging).

I have some more specific comments:

My main problem with the study is that it is mostly descriptive: no associations with atrial arrhythmias or outcome were studied. The focus on CRT responders is in my view intrinsically wrong. There is a large group of patients with discordant remodelling (i.e. left atrial reverse remodelling in CRT non-responders) with improved outcome (Mathias et al. JACC 2016, Kloosterman et al. Europace 2016). The conclusions are unclear to me: if LA remodelling is related to LV remodelling why does it need to be assessed? The most interesting group would be the atrial non-responders in the LV responder group. Hypothetically, in this group remodelling has exceeded the "point of no return". The manuscript needs another round of editing to correct grammatical/semantical mistakes (e.g. LA reserve remodelling instead of reverse remodelling in the first sentence of the conclusions).

Response: Thank you for your suggestion. We have now revised the discussion along the lines of your comments.

We have checked the available evidence for isolated and combined LV and LA remodeling and their relationship with clinical outcome but unfortunately the two published papers Mathias et al. JACC 2016) and (Kloosterman et al. Europace 2016) provide no usable data on LA function. Of note, our previous meta-analysis showed that discordance of LA and LV reverse remodelling was not significant (the percentage of mean change of LAVI was related to change of LVESV β = –1.02 (–1.46 to –0.58), p < 0.001) and LVEF (β = 2.02 (1.86–4.58), p = 0.001, decreased LVESV and/or in- creased LVEF were associated with LAVI reduction).

Reviewer 3 Report

Authors present a meta-analysis of LA function in CRT patients. Topic is relevant. Methods are appropriate. Language and style are sound.

However, the reviewer wonders about the shortness of the discussion section, which only counts 9 lines (!). If the results don't need to be discussed, they are probably not worth being presented. The manuscript would surely deserve relevant extension of this section, discussing preliminary data, clinical implications and future directions. 

Author Response

Reviewer 3

Authors present a meta-analysis of LA function in CRT patients. Topic is relevant. Methods are appropriate. Language and style are sound.

However, the reviewer wonders about the shortness of the discussion section, which only counts 9 lines (!). If the results don't need to be discussed, they are probably not worth being presented. The manuscript would surely deserve relevant extension of this section, discussing preliminary data, clinical implications and future directions. 

Response: Thank you for the suggestion.  We have revised the discussion along the lines of your comments

Round 2

Reviewer 2 Report

I thank the authors for condering my earlier comments.

Even though they are not considering Kloosterman et al. and Mathias et al. for their analysis, I believe concordant and discordant remodeling should be discussed with at least a few sentences in the discussion section.

The authors refer to their own work too extensively, reference 1 and 37 should be deleted and reference 5 and 39 are identical.

Author Response

Response:

Thank you for your great suggestions.  

We removed the references and now we have revised the discussion along the lines of your comments.

We have checked again the available evidence but unfortunately the two published papers Mathias et al. JACC 2016) and (Kloosterman et al. Europace 2016) provide no usable data on LA function.

Reviewer 3 Report

Authors extended discussion section, which is appreciated.

Author Response

Thanks for positive evaluation.